# Phylogenetic and Genetic Variation Analysis of Porcine Epidemic Diarrhea Virus in East Central China during 2020–2023

**DOI:** 10.3390/ani14152185

**Published:** 2024-07-26

**Authors:** Liumei Sun, Duo Li, Caijie Yan, Chengyue Wu, Feng Han, Zongyi Bo, Manman Shen, Yiwei Sun, Liyan Wang, Haoqin Zheng, Mengdong Wang, Zhendong Zhang

**Affiliations:** 1Jiangsu Key Laboratory of Sericultural and Animal Biotechnology, School of Biotechnology, Jiangsu University of Science and Technology, Zhenjiang 212100, China; sun.liumei@163.com (L.S.); liduo649@163.com (D.L.); yancaijie120724@163.com (C.Y.); wuchengyue0817@126.com (C.W.); hanfeng62122024@163.com (F.H.); shenman2005@163.com (M.S.); sunyiwei0224@163.com (Y.S.); stef1214@163.com (L.W.); 13790824740@163.com (H.Z.); mengdong@cau.edu.cn (M.W.); 2Key Laboratory of Silkworm and Mulberry Genetic Improvement, Ministry of Agriculture and Rural Affairs, Sericultural Scientific Research Center, Chinese Academy of Agricultural Sciences, Zhenjiang 212100, China; 3Joint International Research Laboratory of Agriculture and Agri-Product Safety, The Ministry of Education of China, Yangzhou University, Yangzhou 225009, China; zybo@yzu.edu.cn; 4Jiangsu Co-Innovation Center for the Prevention and Control of Important Animal Infectious Disease and Zoonoses, College of Veterinary Medicine, Yangzhou University, Yangzhou 225009, China

**Keywords:** PEDV, Spike gene, phylogenetic analysis, recombination analysis, antigenic analysis

## Abstract

**Simple Summary:**

Simple Summary: Porcine epidemic diarrhea virus (PEDV) is a significant pathogen that has resulted in substantial economic ramifications within the worldwide swine industry. Since 2010, the emergence of novel variants of PEDV has been ongoing, resulting in frequent reclassification of PEDV strains in China. In this investigation, we found the emergence of nine variants in East Central China during 2020–2023. The S protein of three variants was likely derived from recombination of parental variants with a donor variant. There are novel mutations on amino acid 141–148 and these resulted in changes in antigenicity in the three variants. This research has the potential to serve as a basis for the development of a vaccine for PEDV.

**Abstract:**

Porcine epidemic diarrhea virus (PEDV) is a major causative pathogen of a highly contagious, acute enteric viral disease. This study evaluated the emergence of nine variants in Jiangsu and Anhui provinces of China from 2020 to 2023. S gene-based phylogenetic analysis indicated that three variants belong to the G1c subgroup, while the other six strains are clustered within the G2c subgroup. Recombination analyses supported that three variants of the G1c subgroup were likely derived from recombination of parental variants FR0012014 and a donor variant AJ1102. In addition, there are novel mutations on amino acid 141–148 and these likely resulted in changes in antigenicity in the three variants. These results illustrated that the study provides novel insights into the epidemiology, evolution, and transmission of PEDV in China.

## 1. Introduction

Porcine Epidemic Diarrhea Virus (PEDV) is a direct causative pathogen of diarrheic diseases in piglets belonging to the genus Alphacoronavirus in the family Coronaviridae [1]. PEDV can infect all ages of pigs and lead to up to 100% mortality in neonatal suckling piglets within 7 days of age when infected with the virulent PEDV strains [2,3]. With the continuous emergence of PEDV variant strains, the difficulty of PEDV prevention and control has increased, which has caused huge economic losses in the global pig industry [4].

The PEDV genome is a positive-sense, single-stranded RNA (ssRNA) with no segments, encoding nonstructural protein nsp1–16, structural protein S protein, accessory protein ORF3, and structural proteins E, M, and N from the 5′ end to the 3′ end in order [5]. The PEDV virion consists of a lipoprotein envelope and a nucleocapsid. The lipoprotein envelope includes S protein (spike protein), M protein (membrane protein), and E protein (envelope protein), which are located outside the nucleocapsid; the nucleocapsid includes N protein (nucleocapsid protein) and viral genomic RNA [6]. Among these proteins, the S protein is the major antigen of PEDV, which is critical for virus adsorption, receptor binding, membrane fusion, and entry [7]. The S gene of PEDV is prone to recombination, insertion, and deletion mutations and has high variability, which is crucial in virus pathogenicity, transmission, and evolution [8]. Therefore, the S gene is generally known as the indicator of PEDV genetic evolution [9].

Due to the continuous variation of the genome, the classification of the PEDV genotype is constantly changing [10,11,12,13,14]. The classification of the PEDV genotype can be based on the homology of the genome, ORF1a, ORF1b, S, ORF3, E, M, and N genes [15,16,17]. Among them, the evolution and classification of PEDV have been studied most by the sequence of PEDV full-length genomes and Spike genes [6,18,19,20,21]. Initially, according to the homology of the S gene, PEDV was classified into four subtypes: Gla, G1b, G2a, and G2b [22]. Recently, PEDV was classified into six subtypes: G1a, G1b, G1c(S-INDEL), G2a, G2b, and G2c based on the homology of the S gene [23,24].

To research the diversity of PEDV strains in some areas of Jiangsu and Anhui provinces in China, fecal samples of pigs with diarrhea were collected for detection. The S genes were cloned and sequenced, and S gene-based phylogenetic analysis was carried out. In addition, the S gene-based recombination, alignment of amino acids, and antigenic index were analyzed. This study provides evidence of the genetic diversity of PEDV and has significant implications for the diversity and evolution of PEDV.

## 2. Materials and Methods

### 2.1. Clinical Sample Information

Our researchers collected fecal samples from six cities of Jiangsu and Anhui provinces (Suqian, Xuzhou, Yancheng, Nantong, Fuyang, and Haozhou) in China from 2020 to 2023. A total of 115 fecal samples were randomly collected from diseased piglets with diarrheic symptoms from 10 farms. In this study, nine representative PEDV-positive samples were used for further evaluation. The fecal samples were collected and placed in autoclaved collection tubes. After the Dulbecco’s Modified Eagle’s Medium (DMEM) was dissolved, the supernatant was centrifuged and filtered using a sterile filter with a pore size of 0.45 μm. The fecal filtrate was obtained for the extraction of viral RNA.

### 2.2. Primer Design

Primers for detecting PEDV (PEDV-SF, PEDV-SR) were designed according to the relatively conserved region sequence of the S gene. Four pairs of amplification primers (PEDV-S1F/PEDV-S1R, PEDV-S2F/PEDV-S2R, PEDV-S3F/PEDV-S3R, PEDV-S4F/PEDV-S4R) were used to amplify the S gene in four fragments. Repeat-fragment regions were set up between each fragment, which can reduce the probability of replication error when the gene is amplified and sequenced. The detection primers and amplification primers of the S gene are shown in Table 1.

### 2.3. Real-Time PCR Detection and S Gene Sequencing

Total RNA was isolated from the virus solution using TRIzol reagent (purchased from Thermo Fisher Scientific, Waltham, USA) following the manufacturer’s instructions. The RNA was stored at −80 °C for backup. The cDNA was synthesized using a HiScriptII^®^ Reverse Transcriptase kit (purchased from Vazyme Biotech Co., Ltd., Nanjing, China) following the manufacturer’s instructions. Primers were designed by using SnapGene software. The cDNA was used as the template, PCR amplification was performed using detection primers (PEDV-SF/PEDV-SR), and PCR products were identified by agarose gel electrophoresis. The cDNA of positive samples was used as the template, and PCR amplification was performed using amplification primers (PEDV-S1F/PEDV-S1R, PEDV-S2F/PEDV-S2R, PEDV-S3F/PEDV-S3R, PEDV-S4F/PEDV-S4R) and PrimeSTAR^®^ Max DNA Polymerase (purchased from TaKaRa Biotechnology Co., Ltd., Dalian, China). The size of the PCR product was identified by agarose gel electrophoresis, and a gel extraction kit (purchased from Vazyme Biotech Co., Ltd., Nanjing, China) was used to recover PCR products. PCR products were sent to company (Beijing Tsingke Biotech Co., Ltd., Beijing, China) for sequencing.

### 2.4. Phylogenetic Analyses

All S protein sequences from the sample strains and downloaded from GenBank strains (Table 2) were analyzed by clustalx1.83. Phylogenetic analysis based on the S gene was carried out using the neighbor-joining method in the MEGA-X v.10.1.8 program. The robustness of the phylogenetic tree was evaluated by bootstrapping using 1000 replicates.

### 2.5. Recombinant Analyses

First, all the S sequence data in this study were screened using Recombination Detection Program version 4 (RDP4); the set for recombination analyzed using RDP, GENECONV, MaxChi, Chimaera, and 3Seq; followed by secondary scanning and recombination using SiScan and BootScan. Sequences with significant signals for recombination determined by more than two methods were analyzed in greater detail. Nucleotide sequence similarity of all the S sequences in this study was detected by SimPlot v.3.5.1 [25], with a sliding window size of 500 bp, step size of 100 nucleotides, and 1000 bootstrap replicates, using gap-stripped alignments and the F84 (ML) distance model.

## 3. Results

### 3.1. Identity and Homology Analysis of S Gene Sequence

We collected fecal samples from pigs with diarrhea in 10 farms; RT-PCR results showed that of the 115 samples from pigs with diarrhea tested, 21 (about 18.26%) were PEDV positive. The positive samples’ S1, S2, S3, and S4 gene fragments were amplified with four pairs of S gene segmentation primers, and four target bands were obtained, the size consistent with the expectation (Figure 1).

The target fragments of 21 positive samples were recovered and sent to the company for sequencing. The 21 representative positive samples were sequenced by the whole S gene, and the four amplified fragments of the S gene of the 21 positive samples collected in this study were spliced by SeqMan v.11.2 software. The sequencing results showed that we obtained nine different S gene sequences of 21 positive samples; we named them JSnt2020, JSsq2021, JSxz2021, JSyc2021, AHbz2022, AHfy2023, JSxz2023, AHbz2023-1, and AHbz2023-2 strains. The AHbz2023-1 and AHbz2023-2 strains were detected in the same pig farm, and the other strains were detected in different pig farms, respectively. The results showed that the nucleotide lengths of the JSnt2020, JSsq2021, JSxz2021, JSyc2021, AHbz2022, AHfy2023, JSxz2023, AHbz2023-1, and AHbz2023-2 strains were 4149 bp, 4161 bp, 4161 bp, 4155 bp, 4161 bp, 4161 bp, 4161 bp, 4149 bp, and 4149 bp, respectively. The nucleotide and amino acid homology of all S genes of nine strains collected in this study were 95.7–99.7% and 95.1–99.5%, respectively. The nucleotide and amino acid homology of nine strains collected in this study compared with reference strains CV777 (G1a), Vaccine-CV777 (G1b), CH/HNBR/01/2021 (G1c), AH2012 (G2a), AJ1102 (G2b), CHN-SC2021 (G2c) are shown in Table 3. The results showed that the nucleotide sequences of the nine strains collected in this study were different from the typical strains (CV777 and Vaccine-CV777) and the variant strains (AH2012 and AJ1102), and the nucleotide sequences of the nine strains collected in this study were similar to those of the domestic popular strains in recent years in China.

### 3.2. Phylogenetic Analysis of PEDV Based on Nucleotide Sequences of the S Gene

Phylogenetic analysis was performed on the S genes of nine detected strains and different regional strains and vaccine strains. The results showed that the evolutionary tree is mainly divided into two large branches, namely the G1 genotype and G2 genotype, among which the G1 genotype is further divided into G1a, G1b, and G1c subtypes, and the *G2* gene group is further divided into G2a, G2b, and G2c subtypes (Figure 2). In this study, the JSyc2021, JSxz2021, JSsq2021, JSxz2023, AHbz2022, and AHfy2023 strains were closely related to reference strain CHN-SC2021 found in China in 2021, and belong to the G2c subtype. The JSnt2020, AHbz2023-1, and AHbz2023-2 strains were closely related to reference strain CH/HNBR/01/2021 found in China in 2021, and belong to the G1c subtype. These nine strains collected in this study were distant from the classical strains (CV777, DR13, and SD-M) and the variant strains (AH2012 and AJ1102), which were prevalent in China in earlier years.

### 3.3. Recombination Analysis of PEDV Based on Nucleotide Sequences of the S Gene

To determine whether the detected strains were potential recombinants from reference strains, the aligned S genes were all scanned for recombination events using seven algorithms (RDP, GENECONV, BootScan, Maxchi, Chimaera, SiScan, and 3Seq) implemented in RDPv.4.39 [26]. The RDP4 results revealed that three GI-c genogroup strains (JSnt2020, AHbz2023-1, and AHbz2023-2) were probably generated via inter-genogroup recombination (Figure 3). To further evaluate recombination events and determine parents, we performed S gene similarity comparisons between the JSnt2020, AHbz2023-1, and AHbz2023-2 strains and other subgroups strains with SimPlot v.3.5.1, as demonstrated in Figure 3. Results showed that about 710–1190 bp of the S gene in the AHbz2023-1, AHbz2023-2, and JSnt2020 strains had the highest similarity with the FR0012014 strain, and the remaining S gene had the highest similarity with AJ1102 strain. The recombination breakpoints were found to be located within the nucleotides 719–1191, 712–1194, or 712–1022 of S genes of the JSnt2020, AHbz2023-1, and AHbz2023-2 strains. The result indicated that the JSnt2020, AHbz2023-1, and AHbz2023-2 strains were recombinant strains originating from the FR0012014 strain and virulent strain AJ1102. The PEDV S gene recombinants have three major fragments; at least two cross-overs are likely required to generate such recombinants.

### 3.4. Comparative Analysis of Amino Acid Sequences of S Protein

The S protein is critical for virus entry into cells and induction of the host immune response because it is bound to cell receptors and owns four B-cell epitopes [27]. However, the S gene of PEDV is prone to mutation, which accelerates the evolution of the virus [28,29]. To elucidate the S gene genetic identity of the detected strains, the deduced amino acid sequences of nine detected strains were compared with 26 historic representative reference strains from each subgroup. A sequence alignment showed that six out of nine PEDV detected strains (JSsq2021, JSxz2021, JSyc2021, AHbz2022, AHfy2023, and JSxz2023) have the same insertions (“G56ENQ59” and “N144”) and deletions (“D164G165”), similar to other G2 variants. In contrast, the JSnt2020, AHbz2023-1, and AHbz2023-2 strains did not have these insertions and deletions, similar to other G1 strains (Figure 4). However, these three strains have the aa mutation and delete mutation in 141–148 aa compared to other subtypes of viruses. In addition, we compared all neutralizing epitope mutations, including COE (499–638 aa), SS2 (748–755 aa), SS6 (764–771 aa) and 2C10 (1368–1374 aa). The results showed that one aa mutation was observed in the COE (499–638 aa) neutralizing epitopes of the JSnt2020 (D575E), AHfy2023 (T637M), JSyc2021 (S571P), JSxz2021 (T553K), and JSsq2021 (S571Y) strains; no aa mutation was found in other neutralizing epitopes in detected strains. Interestingly, compared with other subgroups, the JSnt2020, AHbz2023-1, and AHbz2023-2 strains, belonging to G1c group, contained 13 distinct patterns of aa mutations (S28L, I71L, N121G, I123V, T141S, V142S, N143G, T148S, I168V, V170I, T241I, M309I, and L1004M) and one deletion at position 145.

### 3.5. Different Antigenic Index of PEDV S Protein

The S protein is the major antigenic protein that can induce the neutralizing antibody against PEDV [24]. To detect whether there was antigenic change in the novel detected strains, the antigenic index of the S proteins of nine detected stains and the representative strain (G1a-CV777, G1b-CV777 Vaccine, G1c-ZL29, G2a-AH2012, G2b-AJ1102, and G2c-CHN-SC2021) of each genotype were analyzed using the Jameson–Wolf algorithm method in DNASTAR v.7.1 software. As shown in Figure 5A, compared with the representative strains, the antigenic index of the novel detected strains JSnt2020, AHbz2023-1, and AHbz2023-2 was similar to those of the G1 group strains in region (120–280 aa), and the antigenic index of the novel detected strains JSsq2021, JSxz2021, JSyc2021, AHbz2022, AHfy2023, and JSxz2023 was similar to those of the G2 group strains in region (120–280 aa). Additionally, compared with the G1 group strains, the G2 group strains had a different antigenic index in the region (120–280 aa). These findings might help explain why the vaccines of G1 group strains do not provide optimal protection against the G2 group strains of PEDV. Furthermore, the JSnt2020, AHbz2023-1, and AHbz2023-2 strains had a different antigenic index in the region (120–150 aa). These three strains have the aa mutation and delete mutation in 141–148 aa compared to other subtypes of viruses. The mutations in these amino acid sites were suspected to affect their antigenicity. Therefore, the S protein structure of the JSnt2020, AHbz2023-1, and AHbz2023-2 strains were predicted using SWISS-MODEL according to the structure of the PEDV in the PDB database (accession code 7w6m). Structure prediction showed that the 141–148 aa is located between two domains at the surface of the S protein (Figure 5B). The amino acids mutation in 141–148 aa may alter the formation of hydrogen bonds, which may affect the antigenicity of the S protein.

## 4. Discussion

Since variant strains emerged in late 2010, PEDV has led to heavy mortality and serious threats to the global swine industry [30]. Due to the difference between clinical vaccine strains and epidemic strains, existing vaccines cannot effectively prevent the epidemic of PEDV [14]. Therefore, timely monitoring of PEDV prevalence and analysis of mutation of the S gene sequence can provide the basis for the development of efficient vaccines and guide the effective prevention and control of PEDV.

This study characterized the PEDV variants circulating in piggery in Jiangsu and An-hui provinces of China in recent years. In addition, novel substitutions, deletions, and insertions could be detected in the 2020–2023 PEDV strains. Remarkably, inter-subgroup recombination events were detected in PEDV strains, supporting that PEDV cross-over events have occurred in piggery to generate novel recombinants. PEDV S gene recombinants have three major fragments; at least two cross-overs are likely required to generate such recombinants. For instance, imagine the first cross-over combines the beginning of the FR0012014 strain’s sequence with the middle of the AJ1102’s sequence. Then, a second cross-over might combine the end of the FR0012014 strain’s sequence with the remaining part of the AJ1102’s sequence, thus creating three distinct fragments. Finally, we found that detected strains belonging to different subgroups exhibited distinct variation patterns on the antigenic index in the N terminal domain of S protein. These data describe the diversity of PEDV and the sequence characteristics of the S gene, providing basic data for enriching the epidemiological data of PEDV.

With the continuous emergence of PEDV variant strains, the classification of PEDV genotypes is increased. PEDV strains are usually classified into G1 and G2 genotypes based on the homology of the S gene of PEDV [6,14]. Genotype G1 strains emerged in the 1970s, such as CV777, the first isolated PEDV strain in the world [31]. Since 2010, the genotype G2 strains have been prevalent globally [3]. As genogroups G1 and G2 further evolved, they were divided into many different subgroups. In 2013, when the classification of PEDV genogroups had just started, PEDV was uniformly divided into three groups: group1, group2, and group3 [32,33]. During 2013–2018, the PEDV genogroups were divided into G1 and G2; the G1 genogroups were further divided into two sub-genogroups, G1a and G1b; and the G2 genotype was further divided into two sub-genogroups, G2a and G2b [6,22,34]. After 2018, the third important subtype, GII-c, was added to the GII group, which was a kind of S-INDEL strain produced by recombination of the subgroups GI-a and GII-a based on the nucleotide sequence of the S gene [15,35]. Recently, a third important subtype G1c was added to the G1 group, which includes S-INDEL strains such as USA/Iowa106/2013, MYZ-1/JPN/2013, and some strains isolated in southwest China during 2015–2018 [23]. In this study, according to the above classification methods, PEDV genotypes were divided into six subtypes: G1a, G1b, G1c, G2a, G2b, and G2c, based on the nucleotide sequence of the S gene. The phylogenetic analysis results showed that JSnt2020, AHbz2023-1, and AHbz2023-2 belong to the G1c subgroup, while JSyc2021, JSxz2021, JSsq2021, JSxz2023, AHbz2022, AHfy2023 are clustered within the G2c subgroup.

Considering that some strains in the G1c subgroup may be recombined from other strains, for example, the ZL29 strain may have been recombined from the G1a and G2a subgroups [15], we analyzed whether the nine detected strains were recombined from other reference strains. To detect if any strains were recombined from other reference strains, the RDP4 software was used to analyze all strains in this study [36,37]. According to the analysis, the three detected strains JSnt2020, AHbz2023-1, and AHbz2023-2 may be recombined from the G1c (FR0012014 strain) and G2b (AJ1102 strain) subgroup strains. To further evaluate the possibility of a recombination event, SimPlot 3.5.1 software was used to analyze the performed S gene similarity comparisons between the JSnt2020, AHbz2023-1, and AHbz2023-2 strains and other subtypes strain [38]. According to the analysis results of the RPD4 and SimPlot software, and referring to previous data [15], we concluded that the S gene sequences of the three detected strains (JSnt2020, AHbz2023-1, and AHbz2023-2) may be recombined from the S gene sequences of the G1c and G2b subgroup strains. The intermediate sequence of these three S1 genes may be derived from the G1c subgroup strains, while the rest are derived from the G2b subgroup strain.

Considering the importance of the S protein to the PEDV virus, we compared the amino acid sequences of the S protein of the nine detected strains with the representative strains in each subgroup (Figure 4). Results showed that G2 genotype strains have the same insertions (“G56ENQ59” and “N144”) and deletions (“D164G165”) compared to G1 genotype stains. In addition, compared to other strains, JSnt2020 and AHfy2023 strains have aa mutations observed in the COE (499–638 aa) neutralizing epitopes [39]. This means that existing vaccines may not prevent the spread of these two strains [40,41,42].

To explore if the amino acid change will influence the antigenicity of PEDV, antigenic index analysis of S protein was performed using the Jameson–Wolf algorithm method in DNASTAR software [43]. The results showed that the antigenicity was very different between the G1 to G2 genotypes in the N terminal of S protein. In addition, the antigenicity was different in the JSnt2020, AHbz2023-1, and AHbz2023-2 compared with other reference strains in 135–150 aa of S protein (in Figure 5). That means the aa mutations and delete mutation in 141–148 aa of the S proteins of these three strains (in Figure 4) influenced their antigenicity.

## 5. Conclusions

In summary, we found that two subgroups of PEDV strains, the G1c subgroup and the G2c subgroup, were prevalent in Jiangsu and Anhui provinces of China during 2020–2023. In addition, we also detected inter-genogroup recombination events involved in the evolution of three detected G1c strains (JSnt2020, AHbz2023-1, and AHbz2023-2). The prevailing G1c and G2c strains exhibited distinct variation patterns in the amino acid sequence and the antigenic index in the N terminal domain of the S protein. These findings will help to understand the prevalence, genetic characteristics, and evolutions of circulating PEDV strains in China.

## Figures and Tables

**Figure 1 animals-14-02185-f001:**
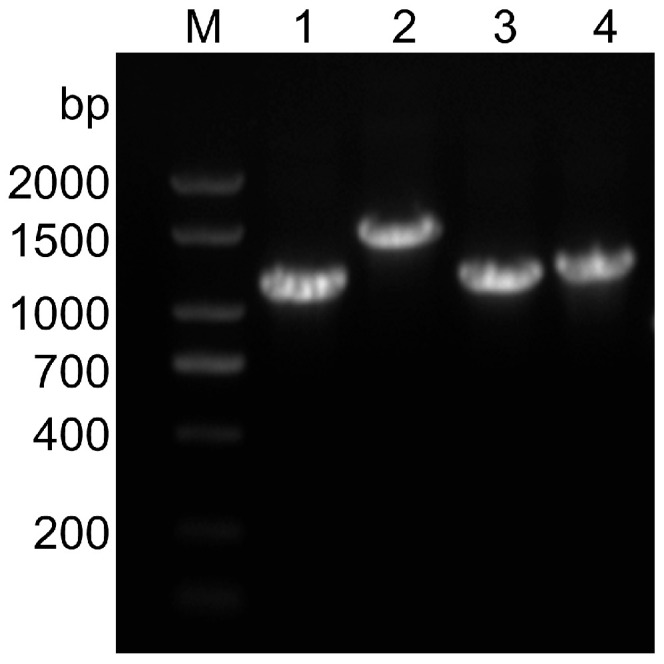
The electrophoresis results in amplified fragments of the S gene. Lane M, 2000 bp DNA ladder; Lane 1–4: S1, S2, S3, S4 gene sequence, constituting S gene full-length sequence.

**Figure 2 animals-14-02185-f002:**
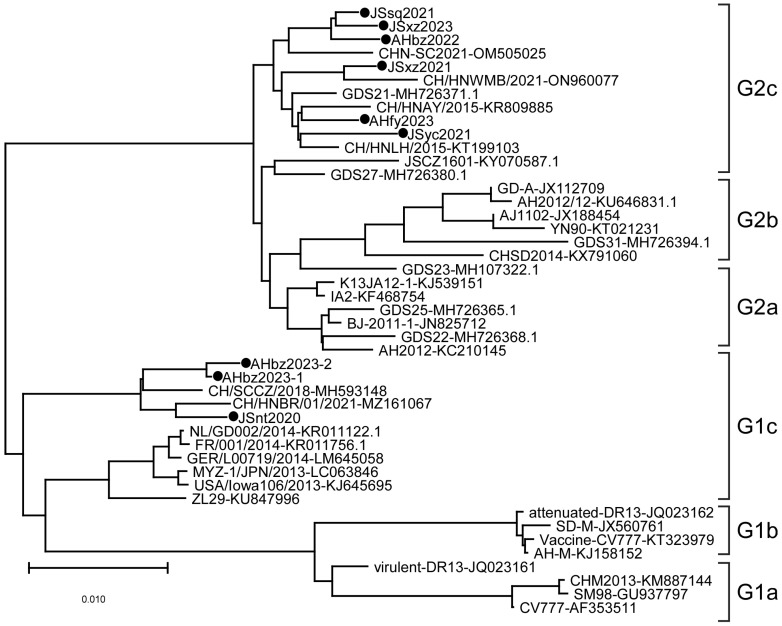
Phylogenetic analysis of PEDV based on nucleotide sequences of the S gene. The phylogenetic tree was constructed with MEGA-X v.10.1.8 software using the neighbor-joining method. Bootstrap analysis was set in 1000 replicates, with a value > 70%, to assess the significance of the tree topology. The information on reference strains is provided in Table 2. “●” indicates the strains detected in this study.

**Figure 3 animals-14-02185-f003:**
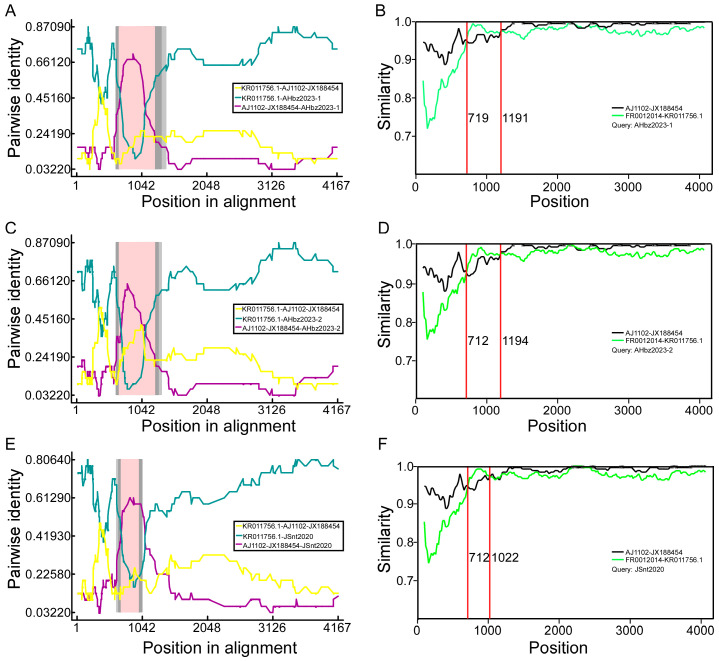
Recombination analysis of S gene of AHbz2023-1, AHbz2023-2, and JSnt2020. The RDP plot clearly illustrates the recombination events detected in AHbz2023-1 (**A**), AHbz2023-2 (**C**), and JSnt2020 (**E**). The similarity of the S gene of AHbz2023-1 (**B**), AHbz2023-2 (**D**), and JSnt2020 (**F**) compared with FR0012014 and AJ1102 is shown.

**Figure 4 animals-14-02185-f004:**
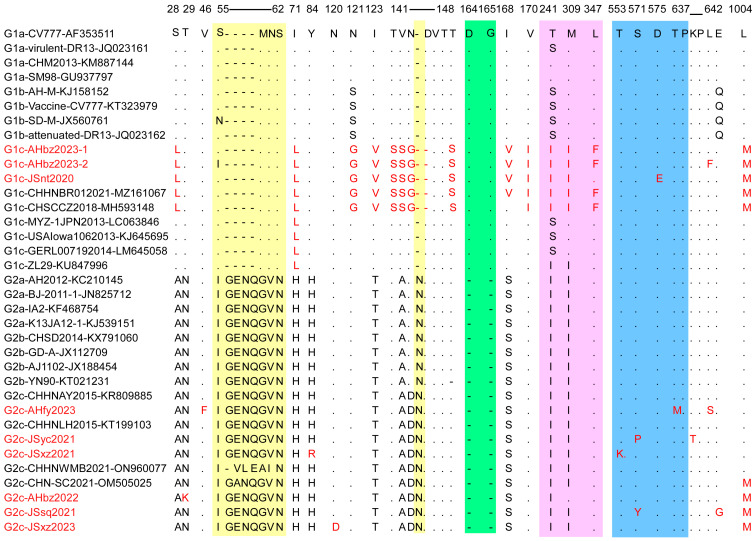
Alignment of amino acid sequences of S proteins of PEDV detected strains and reference strains. The vaccine strain CV777 (GenBank accession no. AF353511) was set as a reference. The amino acid insertions are colored on a yellow background. The amino acid deletions are marked in a green background. The amino acid mutations in the acquired region (710–1190 bp) are shown in pink. The amino acid mutations in the COE (499–638 aa) region are shown in blue. The amino acid mutations and detected strains are highlighted in red.

**Figure 5 animals-14-02185-f005:**
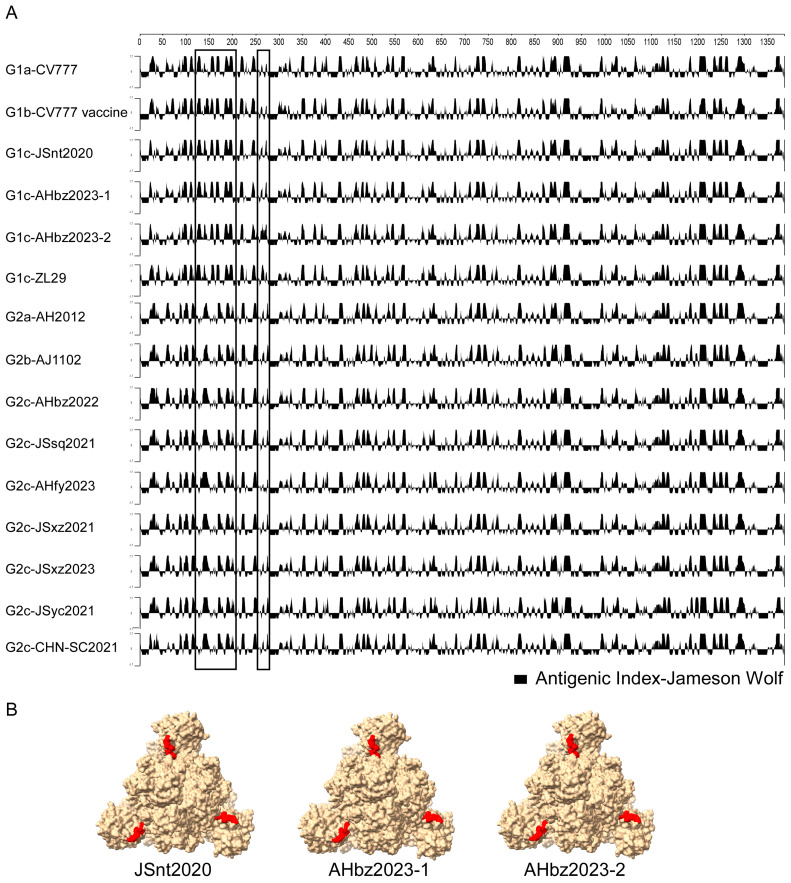
Different antigenic indices of PEDV S protein. (**A**) Antigenic index plots of the amino acid sequences of S protein. The antigenic index plots were calculated using the Protean of DNASTAR Lasergene v.7.1 software under the Jameson–Wolf algorithm. The graphic above zero represents the predictive antigenic sites, and the antigenic discrepancy in the detected strains was labeled with a rectangle. (**B**) The predicted three-dimensional (3-D) modeling of the S protein of JSnt2020, AHbz2023-1, and AHbz2023-2 strains.

**Table 1 animals-14-02185-t001:** The primer information of the S gene.

Name	Sequence (5′-3′)	Product Length/bp
PEDV-SF	CATACAGTACTTGTACCGGGTGA	384
PEDV-SR	AGACTTGGACCATTTCTA	
PEDV-S1F	GAAGAATGGTAAGTTGCTAGTG	1100
PEDV-S1R	GAAGTACAATTGAGCCTTCAGC	
PEDV-S2F	GGCCATTCCTAAGATTTATGG	1400
PEDV-S2R	CACCTATGTTACTATACACC	
PEDV-S3F	GATGATGATATAGTGGGTG	1200
PEDV-S3R	CTGTACCAGAGAGAAAATG	
PEDV-S4F	ACTAAGTATACTGAGGTTC	1000
PEDV-S4R	TGGAACTACATTGAGCTC	

**Table 2 animals-14-02185-t002:** The S gene of information of PEDV detected and reference strains.

	Name of Strain	Access. No.	Location	Date
1	SM98	GU937797	South Korea	2010
2	CHM2013	KM887144	China	2013
3	CV777	AF353511	Switzerland	2001
4	Virulent-DR13	JQ023161	South Korea	2011
5	Attenuated-DR13	JQ023162	South Korea	2011
6	SD-M	JX560761	China	2012
7	Vaccine-CV777	KT323979	China	2015
8	AH-M	KJ158152	China	2011
9	ZL29	KU847996	China	2016
10	GER/L00719/2014	LM645058	Germany	2014
11	USA/Iowa106/2013	KJ645695	USA	2013
12	MYZ-1/JPN/2013	LC063846	Japan	2013
13	CH/SCCZ/2018	MH593148	China	2018
14	CH/HNBR/01/2021	MZ161067	China	2021
15	K13JA12-1	KJ539151	South Korea	2013
16	IA2	KF468754	USA	2014
17	BJ-2011-1	JN825712	China	2011
18	AH2012	KC210145	China	2012
19	YN90	KTO21231	China	2014
20	AJ1102	JX188454	China	2012
21	GD-A	JX112709	China	2012
22	CHSD2014	KX791060	China	2014
23	CHN-SC2021	OM505025	China	2021
24	CH/HNWMB/2021	ON960077	China	2021
25	CH/HNLH/2015	KT199103	China	2015
26	CH/HNAY/2015	KR809885	China	2015
27	JSnt2020	OR808012	China (Jiangsu)	2020
28	JSyc2021	OR808013	China (Jiangsu)	2021
29	JSxz2021	OR808014	China (Jiangsu)	2021
30	JSsq2021	OR808015	China (Jiangsu)	2021
31	AHbz2022	OR808016	China (Anhui)	2022
32	JSxz2023	OR808017	China (Jiangsu)	2023
33	AHfy2023	OR808018	China (Anhui)	2023
34	AHbz2023-1	OR808019	China (Anhui)	2023
35	AHbz2023-2	OR808020	China (Anhui)	2023
36	GDS21	MH726371	China (Anhui)	2014
37	GD22	MH726368	China (Jiangsu)	2012
38	GDS23	MH107322	China (Jiangsu)	2012
39	GDS25	MH726365	China (Jiangsu)	2013
40	GDS27	MH726380	China (Anhui)	2014
41	GDS31	MH726394	China (Jiangsu)	2015
42	JSS01	KU646831	China (Anhui)	2012
43	JSS12	KY070587	China (Jiangsu)	2016
44	FR0012014	KR011756.1	France	2014
45	NLGD0022014	KR011122.1	The Netherlands	2014

**Table 3 animals-14-02185-t003:** Homology analysis of the S gene of nine PEDV strains and six reference strains.

PEDV Strains (Subgroup)	Percentage of Nucleotide (Amino Acid) Identity (%)					
	CV777	Vaccine-CV777	CH/HNBR/01/2021	AH2012	AJ1102	CHN-SC2021
JSyc2021	94.0	93.7	95.7	98.1	97.7	98.5
(G2c)	(93.3)	(92.7)	(95.1)	(97.9)	(98.2)	(98.1)
JSxz2021	93.9	93.6	95.8	98.1	97.3	98.5
(G2c)	(93.6)	(93.0)	(95.7)	(98.4)	(98.1)	(98.6)
JSsq2021	94.1	93.8	96.5	98.3	97.5	98.9
(G2c)	(93.6)	(92.9)	(95.7)	(98.3)	(98.1)	(98.6)
AHbz2022	94.1	93.8	96.3	98.2	97.4	98.8
(G2c)	(93.8)	(93.0)	(95.7)	(98.1)	(98.0)	(98.3)
AHfy2023	94.0	93.8	96.1	98.3	97.4	98.7
(G2c)	(93.2)	(92.6)	(95.4)	(98.1)	(97.8)	(98.2)
JSxz2023	94.1	93.7	96.4	98.3	97.4	98.7
(G2c)	(93.7)	(92.9)	(95.9)	(98.4)	(98.2)	(98.7)
JSnt2020	94.9	94.9	99.3	95.9	95.1	96.3
(G1c)	(94.9)	(94.4)	(99.1)	(95.6)	(95.4)	(95.7)
AHbz2023-1	95.0	95.2	99.0	96.2	95.2	96.5
(G1c)	(95.0)	(94.5)	(98.9)	(95.8)	(95.3)	(95.9)
AHbz2023-2	94.7	94.9	98.7	96.1	95.1	96.3
(G1c)	(94.6)	(94.1)	(98.4)	(95.4)	(94.9)	(95.5)

## Data Availability

The data that support the findings of this study are available from the corresponding author upon reasonable request.

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
