# Peer review of "Phylogenetic and Genetic Variation Analysis of Porcine Epidemic Diarrhea Virus in East Central China during 2020–2023"

_animals, 2024, doi:10.3390/ani14152185_

Round 1

Reviewer 1 Report

Comments and Suggestions for Authors

This paper enriches our understanding of the epidemiology of PEDV in Jiangsu and Anhui provinces of CHINA.Here are the specific comments:

1 Line 53 and Line 56, there are levant symbol present within the word of “pro-tein” and “adsorp-tion”.Please check the whole article carefully.

2 Part 2.1 requires a description of the number and area of distribution of the clinical samples.

3 Part 2.2,this paper doesn't seem to use Restriction Endonuclease.If it is necessary, please specify which one?

4 The authors used test primers for screening, but there were no screening results.

5 Whether it is necessary to state that the nine sequences originate from different pig farms?

6 Previous studies identified multiple neutralizingepitopes,I suggest that the authors compare all neutralizingepitopes mutations.

Author Response

Overall Statement: This paper enriches our understanding of the epidemiology of PEDV in Jiangsu and Anhui provinces of CHINA. Here are the specific comments:

Comment #1: Line 53 and Line 56, there are levant symbol present within the word of “pro-tein” and “adsorp-tion”.Please check the whole article carefully.

Response: We deeply appreciated the reviewer’s helpful suggestion. The levant symbol present within the word of “pro-tein” and “adsorp-tion” already be removed from these words, and we checked the whole article to avoid this mistake. (Please refer to line 36-38, 47,50,59,268-269, 296,302,306)

Comment #2: Part 2.1 requires a description of the number and area of distribution of the clinical samples.

Response: We deeply appreciated the reviewer’s insightful suggestion and have made a description of the number and area of distribution of the clinical samples. (Please refer to line 71-79)

Comment #3: Part 2.2,this paper doesn't seem to use Restriction Endonuclease.If it is necessary, please specify which one?

Response: Thanks for your helpful suggestion. We are so sorry for that errors, we didn’t use Restriction Endonuclease. We modified this information. (Please refer to line 89-102)

Comment #4: The authors used test primers for screening, but there were no screening results.

Response: Thanks for your helpful suggestion. We added the description of RT-PCR results in line 121-123.

Comment #5: Whether it is necessary to state that the nine sequences originate from different pig farms?

Response: We deeply appreciated the reviewer’s helpful suggestion. we stated that it in line 135-136.

Comment #6: Previous studies identified multiple neutralizingepitopes,I suggest that the authors compare all neutralizingepitopes mutations.

Response: We deeply appreciated the reviewer’s helpful suggestion. We compared all neutralizing epitopes mutations including COE (499-638aa), SS2 (748–755 aa), SS6 (764–771 aa) and 2C10 (1368–1374 aa). And We described the results in line 204-208. 

Reviewer 2 Report

Comments and Suggestions for Authors

The manuscript basically gains the following 3 points:

1. the emergence of 3 variants during 2020-2023.

2. These 3 variants likely derived from recombination of parental (backbone) variants with a donor variants.

3. There are novel mutations on amino acid 141-148 in the 9 analyzed variants.

Therefore, the simple summary and abstract should be rewritten.  They can be shortened extensively.

line 70: fecal samples were collected from normal piglets, so what you are characterizing here may be non-pathogenic variants.  It would be more significant to sample from diarrhea piglets and correlated the disease with antigenicity change.

line 94: the "mutation" is incorrect.  What you tried to avoid is "replication error" by the amplifying enzyme.

Sections 2.2, 2.4 and 2.5 should be combined into one section.

Table 3: mark, on the first column, the three G1c subgroup variants.

line 165: "distant from" is an alternative for "far related".

section 3.3: the recombinants have 3 major fragments, discuss how many cross-over (2? or at least 2) does it take to get such recombinants.

Figure 4: mark, on this figure, where is the patch of amino acids coded by the acquired "710-1190 bp" (line 182). It should be present on 3 variants.

Figure 5: it is better to have three-dimension (3-D) modeling of the S protein in order to show that this 141-149 aa are actually located or exposed at the key surface for antigenicity.

line 244: discuss how many cross-over (2? or at least 2) does it take to get such recombinant.

Author Response

The manuscript basically gains the following 3 points:

  1. the emergence of 3 variants during 2020-2023.
  2. These 3 variants likely derived from recombination of parental (backbone) variants with a donor variants.
  3. There are novel mutations on amino acid 141-148 in the 9 analyzed variants.

Comment #1: Therefore, the simple summary and abstract should be rewritten.  They can be shortened extensively.

Response: We deeply appreciated the reviewer’s helpful suggestion. The simple summary and abstract were rewritten.

Comment #2: line 70: fecal samples were collected from normal piglets, so what you are characterizing here may be non-pathogenic variants.  It would be more significant to sample from diarrhea piglets and correlated the disease with antigenicity change.

Response: We deeply appreciated the reviewer’s helpful suggestion. Our fecal samples were collected from diseased piglets with diarrheic symptoms. we added this information in line 64, 73-75.

Comment #3: line 94: the "mutation" is incorrect.  What you tried to avoid is "replication error" by the amplifying enzyme.

Response: Thanks for your helpful suggestion, the word was changed into " replication error " accordingly. (Please refer to line 84)

Comment #4: Sections 2.2, 2.4 and 2.5 should be combined into one section.

Response: We deeply appreciated the reviewer’s helpful suggestion. As suggested, sections 2.2, 2.4 and 2.5 already be combined into one section. (Please refer to line 88-102)

Comment #5: Table 3: mark, on the first column, the three G1c subgroup variants.

Response: Thanks for your helpful suggestion, we marked the subgroup of variants on the first column in Table 3. (Please refer to line 148-149)

Comment #6: line 165: "distant from" is an alternative for "far related".

Response: Thanks for your helpful suggestion, the word was changed into " distant from " accordingly. (Please refer to line 160)

Comment #7: section 3.3: the recombinants have 3 major fragments, discuss how many cross-over (2? or at least 2) does it take to get such recombinants.

Response: Thanks for your helpful suggestion, we discuss that in line 179-185. (Please refer to line 160)

Comment #8: Figure 4: mark, on this figure, where is the patch of amino acids coded by the acquired "710-1190 bp" (line 182). It should be present on 3 variants.

Response: Thanks for your helpful suggestion, the patch of amino acids coded by the acquired "710-1190 bp" was present on 3 variants. (Please refer to line 215-216)

Comment #9: Figure 5: it is better to have three-dimension (3-D) modeling of the S protein in order to show that this 141-149 aa are actually located or exposed at the key surface for antigenicity.

Response: We deeply appreciated the reviewer’s helpful suggestion. As suggested, hree-dimension (3-D) modeling of the S protein of 3 variants were shown in Figure 5. And we added the description the results in text. (Please refer to line 232-241)

Comment #10: line 244: discuss how many cross-over (2? or at least 2) does it take to get such recombinant.

Response: We deeply appreciated the reviewer’s helpful suggestion. We discussed that in line 263-268 (Please refer to line 263-268)

Round 2

Reviewer 2 Report

Comments and Suggestions for Authors

The R1 of version of this manuscript has improved.  All my concerns have been addressed.